# CMTN-SP: A Novel Coverage-Control Algorithm for Moving-Target Nodes Based on Sensing Probability Model in Sensor Networks

**DOI:** 10.3390/s19020257

**Published:** 2019-01-10

**Authors:** Zeyu Sun, Xiaofei Xing, Ben Yan, Zhiguo Lv

**Affiliations:** 1School of Computer Science and Information Engineering, Luoyang Institute of Science and Technology, Luoyang 471023, China; zysun@lit.edu.cn (Z.S.); yanben@lit.edu.cn (B.Y.); lzg2008@lit.edu.cn (Z.L.); 2State Key Laboratory of Integrated Service Networks, Xidian University, Xi’an 710071, China; 3School of Computer Science and Educational Software, Guangzhou University, Guangzhou 510006, China

**Keywords:** sensor network, network lifetime, network coverage rate, sensing probability model, coverage control

## Abstract

The non-consecutive coverage problem for the target nodes in Sensor Networks could lead to the coverage blind area and a large amount of redundant data, which causes the bottleneck phenomenon for the communication link. A novel Coverage Control Algorithm for Moving Target Nodes Based on Sensing Probability Model (CMTN-SP) is proposed in this work. Firstly, according to the probability theory, we derive the calculation method for the expectation of the coverage quality with multiple joint nodes, which aims to reduce the coverage blind area and improving network coverage rate. Secondly, we employ the dynamic transferring mechanism of the nodes to re-optimize the deployment of the nodes, which alleviates the rapid exhaustion of the proper network energy. Finally, it is verified via the results of the simulation that the network coverage quality could not only be improved by the proposed algorithm, but the proposed algorithm could also effectively curb the rapid exhaustion of the node energy.

## 1. Introduction

The sensor network is a large-scale and self-organized sensing network which could achieve the information world integration and the physical world integration [1,2,3,4,5]. The deployment of this network is achieved by densely and randomly deploying the nodes which are in the monitoring area, while the sensor nodes behavior is mainly characterized by: a certain degree of computation, communication, storage and control ability, which could accomplish the collection, communication, computation, and storage of the information from the physical world [6,7,8,9]. The high-speed visual sensors are adopted by this work. The main characteristics of this sensor are listed below. (1) Data collection unit. The data collection unit is mainly composed by the sensor and the analog/digit converter, where the data collection module is mainly in charge of collecting the mobile data and the analog/digit converter is in charge of converting the collected analog signal into digital signal. (2) Control unit. The control unit is in charge of the running of the entire sensor node, which is normally composed of the micro-processor and the memory. The function of the micro-processor is to perform the real-time processing on the data collected by the node and received from other nodes. On the other hand, the function of memory is to store the temporary data collected by the node and received from other nodes. (3) Communication unit. The wireless communication unit is in charge of the communication between different sensor nodes, exchanging control messages and transcribing data. (4) Power unit. The power unit is in charge of powering the sensor nodes to guarantee the functioning. It is crucial to the secure and reliable working of the entire network. The coverage problem is crucial to the study of the Sensor Network and closely related to the other research fields of Sensor Network, e.g., node positioning, target tracking, node deployment, time synchronization, data fusion, and node energy consumption [10,11,12,13,14]. The three heated topics of the coverage problem in Sensor Network are (1) how to increase the coverage rate which is effective on the area of monitoring, (2) how to curb the rapid energy exhaustion of the nodes to prolong the network lifetime, and (3) how to use the minimal nodes number to satisfy the required coverage rate for the users in the area of monitoring.

In the covering process on the node which is the target node, usually the full coverage over the entire monitoring area is employed [15,16,17]. However, we utilize the partial consecutive coverage, focusing on the target node. That is, we only consecutively cover the moving target node which we focus on. The main reasons behind this are that (1) during the whole operation period of this network, putting all the nodes into work would rapidly exhaust the node energy, and (2) the required coverage degrees of different monitoring areas are different when we re-partition the whole monitoring area. Therefore, the network energy would also quickly run out if we put all the nodes in the monitoring area into work. Finally, (3) During the consecutive coverage process over the moving node, the coverage blind area would exist since some nodes are dead or not awakened by some algorithm. We take the moving target node as an example and give the illustration of the consecutive coverage, which is shown in Figure 1.

Based on the analysis above, this work mainly focuses on the following four aspects.

(1) We establish the sensing model according to the probability model and utilize the normal distribution feature of multiple nodes to derive the expectation of the coverage quality, aiming to have the coverage rate improved in the area of monitoring.

(2) When a certain coverage rate is satisfied, the result in (1) is employed to derive the calculation method for the minimal nodes number in the area which is monitored, and we further introduce the controllable parameters to achieve the coverage which is effective over the target node that is moving.

(3) According to the curbing of the consumption of the energy, we exploit the dynamic transferring mechanism of the nodes to optimize the deployment and re-partitioning of the nodes. Meanwhile, we perform the resource integration over the nodes which are in the monitoring area to achieve the balance of the energy in the whole network.

(4) Via the simulations, we perform comparisons on the coverage rate of the network, network lifetime, and the quantity of the nodes which are survived, which further verifies the effectiveness of the proposed algorithm.

## 2. Materials and Methods

Recently, the network coverage problem has become a heated topic in the study of Sensor Network. Many researchers in the world has conducted some effective research work and got some results. Liu et al. [7] proposed a probabilistic coverage model based on a circle region in 2005. Li et al. [11] proposed deployment entropy to measure the deployment quantitatively. From the results of the deployment entropy, a scheme of activating pre-deployed sensors is provided to meet the goal of waking up sensors as little as possible to obtain a desirable sensing coverage. The work in Reference [18] mainly studied the impacts of the range and maximum of the particle’s flying speed on the network coverage optimization process. Furthermore, the range of the varying network coverage rate with different number of nodes is also determined. A Multiround Distributed Lifetime Coverage Optimization Protocol (MuDiLCO) was proposed in Reference [19]. This protocol first divides the target area into several subareas and then distributes the MuDiLCO protocol to all the nodes which are called the sensor nodes in each subarea. This protocol works when the set of sensor nodes are scheduled and remain active in the sensing period. Therefore, it determines the sensing coverage area. An Optimized Coverage Algorithm in Probability Model (OCPM) was proposed in Reference [20], which performs investigation and analysis on the sensing model to find the membership between the nodes and the target node coverage. Then it conducts the theoretical calculation and derivation on the rate of the network coverage and gives the calculation method for the deviation when the node covers multiple target nodes simultaneously. A Complex Alliance Strategy with Multi-objective Optimization of Coverage (CASMOC) was proposed in Reference [21]. This strategy employs the proportional relationship between the energy transferring functions of the working node and the neighboring nodes. This proportional relationship is applied to the scheduling of the low-energy moving nodes to achieve the whole network energy balance, optimize the resources of the network and improve the network coverage rate. The Reuleaux Geometry Triangle was employed in Reference [22] to design a Probabilistic Detection Algorithm (PDA). This algorithm first divides the monitoring area in geometry according to the users’ requirement on the coverage rate and then performs the dynamic clustering on the nodes according to the sensing ability of the nodes. The coverage which is effective over the whole area being monitored is achieved via the centralized coverage among different clusters. A Convoy Tree based Fuzzy Target Tracking (CTFTT) algorithm was proposed in Reference [23], which employs the tree-like structure to establish a tree structure focusing on the local node and the neighbor node. It does this by deleting the dead nodes off the tree and adding new nodes to the tree to dynamically update the target tree. Meanwhile, the ambiguity sensing model was exploited to extend, contract, and reconstruct the tree, which accomplishes the effective coverage over the monitoring area. A new Distributed Approach Algorithm Based Trust among Sensor Nodes (DTA) was proposed in Reference [24], which utilizes the sensing ability of the nodes and the remained energy to achieve the effective coverage which is effective over the target nodes. The main idea is to divide the randomly deployed nodes into several clusters while each node in the cluster submits its current state information to the cluster head. Then, the working cluster is determined via the comparison on the cluster energy and the state transferring mechanism of the cluster heads, which achieves the coverage over the target node. Vector of Locally Aggregated Descriptors (VLAD) was proposed in Reference [25]. This algorithm employs the redundancy feature of the nodes to determine the coverage rate and uses the controllability, which is adaptable to the variation of the function, to decline the coverage rate, which is redundant. Therefore, the coverage rate of the network is improved. Deployment entropy was proposed in Reference [26], which is to measure the deployment quantitatively. By the results of the deployment entropy, a scheme of activating pre-deployed sensors is provide to meet the goal of waking up sensors as less as possible to obtain a desirable sensing coverage.

For the entire monitoring area or the target nodes, the algorithms mentioned above could improve the network coverage rate. But there are some deficiencies: (1) some algorithms do not consider the change of the network lifetime when they are performed. (2) The network model is ideal and ignores the boundary effect. (3) The state transferring mechanism and the power allocation mechanism of the nodes are not considered in the coverage process, which causes the rapid exhaustion of the network energy. (4) When the data is relayed, the redundant data problem exists and the routing has to be re-chosen.

Therefore, this work is based on the probability theory and employs the membership between the target node and the node to derive the expectation of the joint coverage with multiple nodes. Based on this, we further give the calculation method for the minimal number of required nodes. In addition, we introduce the controllable parameter to accomplish the coverage which is effective over the whole area which is monitoring. In terms of the energy efficiency, the dynamic clustering algorithm is used to perform the re-integration and scheduling of the nodes within the area which is being monitored to achieve the energy balance of the network.

## 3. Problem Description and Analysis

### 3.1. Assumption and Definition

For further analysis of the algorithm which is proposed to achieve the effective coverage, the proposed algorithm satisfies the following assumption:(1)Then sensor nodes are randomly deployed in the area which is being monitored. The initial energy of all nodes is the same. The sensing area is in the shape of a circle [27].(2)The monitoring area, which is in the shape of a square, is large enough and the edge length is far larger than the sensing radius.(3)An arbitrary node could acquire its location information via the positioning algorithms, e.g., RSSI and DTOA [28].(4)The deployments of the nodes are normally distributed and the communication radius is at least twice as large as the sensing radius [29,30,31].

**Definition** **1.**
***(Coverage Region):** The coverage region is defined as the overall area covered by all the deployed sensor nodes.*


**Definition** **2.**
***(Coverage Quality):** Within the area which is being monitored, the quality of the coverage is defined as the ratio of the union set of all the nodes’ coverage area to the monitoring area.*
(1)$$C_{q}=\frac{\underset{i\in 1,2,\cdots N}{\cup }s_{i}}{\mathrm{area}\left(A\right)}$$
*where C_q_ is the coverage quality and area(A) is the size of the monitoring area and s_i_ is the coverage area of node i.*


**Definition** **3.**
***(Coverage Efficiency)**: The coverage efficiency is the definition of the ratio of union set of all the nodes’ coverage area to the summation of all the nodes’ coverage area.*
(2)$$C_{e}=\frac{\underset{i=1,2\cdots N}{\cup }s_{i}}{\sum_{i=1,2\cdots N}s_{i}}$$
*where C_e_ is the coverage efficiency and s_i_ is the coverage area of node i.*


**Definition** **4.**
***(Coverage Evenness):** Normally, the coverage evenness is defined as the standard deviation of the distance from the reference nodes to the nodes which are neighbor nodes.*
(3)$$E_{v}=\frac{1}{N}{\sum_{i=1}^{N}\left[\frac{1}{NE_{i}}\sum_{j=1}^{NE_{i}}\left(D_{\left(i,j\right)}-d_{i}\right)\right]}^{\frac{1}{2}}$$
*where E_v_ is the coverage evenness and N is the quantity of nodes, NE_i_ is the quantity of neighbor nodes for node i, D(i,j) is the distance between node i and node j and d_i_ is the average distance from all the nodes which intersect with node i.*


**Definition** **5.**
***(Network Lifetime):** The lifetime of the network is defined as the duration of time from the initial time to the time that arbitrary sensor node could not cover an arbitrary target node.*


**Definition** **6.**
***(Coverage Time):** The coverage time is defined as the longest time from the start to the completion of coverage among all the nodes.*


### 3.2. Coverage Quality

For the convenience of research, we assume that the monitoring area is square and *N* sensor nodes are randomly deployed. We take an arbitrary sensor node as the reference node, and its sensing radius *R*_0_ obeys the normal distribution *N*(*R*_0_, *δ*^2^). The average sensing radius in the monitoring area for the nodes is *R_a_*. When the user’s required coverage rate and the communication between nodes and their neighboring nodes are guaranteed, we do not achieve the full coverage over the entire monitoring area. Instead, we only perform consecutive coverage over the moving target node and make sure that the coverage rate varies within a reasonable range.

**Theorem** **1.**
*Assuming that the area of a monitoring area is A and the number of working nodes is N, then the expectation of the coverage quality in this area is $E\left(C_{q}\right)=1-{\left(1-\frac{\pi \left(R_{0}^{2}+\delta^{2}\right)}{\mathrm{area}\left(A\right)}\right)}^{N}$.*


**Proof.** Since we deployed the nodes at a certain monitoring area randomly and the sensing radius *R_i_* of an arbitrary node follows the normal distribution, the coverage rate for an arbitrary node *a* within the monitoring area is:(4)$$p_{a}=\frac{1}{\mathrm{area}\left(A\right)}$$And the probability that an arbitrary node *M* within the monitoring area is covered by *R_a_* is:(5)$$p_{M}=\frac{\pi R_{a}^{2}}{\mathrm{area}\left(A\right)}$$We consider *R*_0_ the sensing radius for the considered reference node and *R_a_* is assumed to be the sensing radius of an arbitrary sensor node within this area, according to the probability density function of the normal distribution, thus we have:(6)$$p=\int_{0}^{2R_{0}}p_{M}\frac{1}{\sqrt{2\pi }\delta }\exp\left(-\frac{{\left(R_{a}-R_{0}\right)}^{2}}{2\delta^{2}}\right)dR_{0}$$Set $x=\frac{R_{a}-R_{0}}{\delta }$ and we have:(7)$$p=\frac{\sqrt{\pi }}{\sqrt{2}\mathrm{area}\left(A\right)}\left(\int_{-\frac{R_{0}}{\delta }}^{\frac{R_{0}}{\delta }}x^{2}\delta^{2}\exp\left(-\frac{x^{2}}{2}\right)dx+2\int_{-\frac{R_{0}}{\delta }}^{\frac{R_{0}}{\delta }}xR_{0}\delta \exp\left(-\frac{x^{2}}{2}\right)dx\right)+\frac{\sqrt{\pi }}{\sqrt{2}\mathrm{area}\left(A\right)}\left(\int_{-\frac{R_{0}}{\delta }}^{\frac{R_{0}}{\delta }}R_{0}^{2}\exp\left(-\frac{x^{2}}{2}\right)dx\right)$$Firstly, we perform the first integration:(8)$$\int_{-\frac{R_{0}}{\delta }}^{\frac{R_{0}}{\delta }}x^{2}\delta^{2}\exp\left(-\frac{x^{2}}{2}\right)dx=-\int_{-\frac{R_{0}}{\delta }}^{\frac{R_{0}}{\delta }}\delta^{2}xd\left(e^{-\frac{x^{2}}{2}}\right)$$Calculated by using the step-by-step integration method:(9)$$\int_{-\frac{R_{0}}{\delta }}^{\frac{R_{0}}{\delta }}x^{2}\delta^{2}\exp\left(-\frac{x^{2}}{2}\right)dx={-\delta^{2}x\exp\left(-\frac{x^{2}}{2}\right)|}_{-\frac{R_{0}}{\delta }}^{\frac{R_{0}}{\delta }}+\delta^{2}\int_{-\frac{R_{0}}{\delta }}^{\frac{R_{0}}{\delta }}e^{--\frac{x^{2}}{2}}dx$$
(10)$$\int_{-\frac{R_{0}}{\delta }}^{\frac{R_{0}}{\delta }}x^{2}\delta^{2}\exp\left(-\frac{x^{2}}{2}\right)dx={-\delta^{2}x\exp\left(-\frac{x^{2}}{2}\right)|}_{-\frac{R_{0}}{\delta }}^{\frac{R_{0}}{\delta }}+\delta^{2}\sqrt{2\pi }$$Then we perform the second integration method. According to the symmetry of the function:(11)$$\int_{-\frac{R_{0}}{\delta }}^{\frac{R_{0}}{\delta }}xR_{0}\delta \exp\left(-\frac{x^{2}}{2}\right)dx=0$$Next, we calculate the third integration:(12)$$\int_{-\frac{R_{0}}{\delta }}^{\frac{R_{0}}{\delta }}R_{0}^{2}\exp\left(-\frac{x^{2}}{2}\right)dx=R_{0}^{2}\int_{-\frac{R_{0}}{\delta }}^{\frac{R_{0}}{\delta }}e^{--\frac{x^{2}}{2}}dx$$
(13)$$\int_{-\frac{R_{0}}{\delta }}^{\frac{R_{0}}{\delta }}R_{0}^{2}\exp\left(-\frac{x^{2}}{2}\right)dx=R_{0}^{2}\sqrt{2\pi }$$We further calculate Equation (13) and get the following result:(14)$$p=\frac{\sqrt{\pi }}{\sqrt{2}\mathrm{area}\left(A\right)}\left({-\delta^{2}x\exp\left(-\frac{x^{2}}{2}\right)|}_{-\frac{R_{0}}{\delta }}^{\frac{R_{0}}{\delta }}+\delta^{2}\sqrt{2\pi }+R_{0}^{2}\sqrt{2\pi }\right)=\frac{\pi \left(R_{0}^{2}+\delta^{2}\right)}{\mathrm{area}\left(M\right)}$$Since the working nodes are assumed to be mutually independent, when an arbitrary moving target node *M* which is in the monitoring area is covered jointly by *k* nodes at least, according to the Binomial theorem, we can derive the expectation of the coverage quality:(15)$$E\left(C_{q}\right)=\sum_{i=1}^{k}\left(\begin{array}{l} k\\ i \end{array}\right)p^{i}{\left(1-p\right)}^{k-i}=1-{\left(1-\frac{\pi \left(R_{0}^{2}+\delta^{2}\right)}{\mathrm{area}\left(A\right)}\right)}^{k}$$It is shown via Equation (15) that when the node which is the target is covered jointly by *N* nodes at least, the expectation of the coverage quality is:(16)$$E\left(C_{q}\right)=1-{\left(1-\frac{\pi \left(R_{0}^{2}+\delta^{2}\right)}{\mathrm{area}\left(A\right)}\right)}^{N}$$This completes the proof. □

**Corollary** **1.**
*The minimal number of the nodes which is required to satisfy the expected coverage quality above is: $N=\ln\left(1-\frac{\pi R_{a}^{2}}{\mathrm{area}\left(A\right)}\right)/\ln\left(1-\frac{\pi \left(R_{0}^{2}+\delta^{2}\right)}{\mathrm{area}\left(A\right)}\right)$.*


**Proof.** When an arbitrary moving node *M* which is the target node and is covered by only one sensor node, the expected coverage quality should be no larger than that when the node which is the target is covered by multiple nodes jointly. We assume the average sensing radius is *R_a_* and it is derived from Equation (16) that:(17)$$\frac{\pi R_{a}^{2}}{\mathrm{area}\left(A\right)}\leq 1-{\left(1-\frac{\pi \left(R_{0}^{2}+\delta^{2}\right)}{\mathrm{area}\left(A\right)}\right)}^{N}$$If we take the logarithmic operation on both sides, it is derived that:(18)$$N=\ln\left(1-\frac{\pi R_{a}^{2}}{\mathrm{area}\left(A\right)}\right)/\ln\left(1-\frac{\pi \left(R_{0}^{2}+\delta^{2}\right)}{\mathrm{area}\left(A\right)}\right)$$This completes the proof. □

### 3.3. Redundant Coverage

In order to guarantee the consecutive coverage which is in the monitoring area over the moving target node and reduce the quantity of redundant nodes and redundant data, we analyze the degree of redundant coverage. We assume the monitoring area is a double square where the outer square is not the focusing area for the moving target node, while the inner square is the focusing area for the target node which is moving while the inner square is the focusing area for the target node which is moving. Generally speaking, the existence of other redundant nodes is ignored for investigation convenience. The illustration of the degree of redundant coverage is shown in Figure 2.

**Definition** **7.**
***(Redundant Node):** When the threshold is higher than the Euclidean distance between two arbitrary nodes, this node is defined as the redundant node, i.e., d(s_i_, s_j_) < ε(l).*


**Definition** **8.**
***(Degree of Redundant Coverage):** When node s_i_ and its neighboring node are redundant nodes, the degree of redundant coverage is defined as the ratio of its neighbor’s coverage area to its own coverage area.*


**Theorem** **2.**
*The degree of redundant coverage for arbitrary n neighbor nodes centering an arbitrary node s_i_ and sharing the same sensing radius R_a_ is derived as follows*
$$p_{R}=1-{\left(1-\frac{1}{\pi^{2}}\left(\frac{\pi }{4}-\frac{1}{8}{\left(4-\lambda^{2}\right)}^{\frac{1}{2}}+\frac{\lambda^{2}\arccos\left(\frac{\lambda }{2}\right)}{2}-\frac{1}{16}\lambda^{3}{\left(4-\lambda^{2}\right)}^{\frac{1}{2}}\right)\right)}^{n}$$


**Proof.** Now we consider Figure 2 and assume that ∠BCE = *α* and *λ* is a controllable parameter, i.e., *BC* = *λR_a_*, *λ* ∈ (0, 2]. We further assume that the distance *l* between B and C is a random variable with the following distribution:(19)$$g_{l}\left(x\right)=\frac{2x}{\lambda^{2}R_{0}^{2}}\;0<x\leq \lambda R_{0}$$The size of the overlapped area between these two points is:(20)$$S=\left(2\alpha -\sin2\alpha \right)R_{0}^{2}$$If we set this distance to be BC = *l*, we have:(21)$$dl=2R_{0}^{2}\sin\alpha d\alpha$$After substituting Equation (21) into the probability density function, we have:(22)$$p_{R}^{\prime }=\int_{\frac{\pi }{2}}^{\arccos\left(\frac{\lambda }{2}\right)}\left(2\alpha -\sin2\alpha \right)\left(\frac{4R_{0}^{2}\cos\alpha }{\lambda^{2}}\right)\sin\alpha da$$We further calculate Equation (22) as:(23)$$p_{R}^{\prime }=\frac{1}{\pi }\left(\frac{\pi }{4}-\frac{\lambda }{8}{\left(4-\lambda^{2}\right)}^{\frac{1}{2}}+\frac{\lambda^{2}\arccos\left(\frac{\lambda }{2}\right)}{2}-\frac{1}{16}\lambda^{3}{\left(4-\lambda^{2}\right)}^{\frac{1}{2}}\right)R_{0}^{2}$$According to the Definition 7 and Definition 8, the degree of redundant coverage for the neighbor node of an arbitrary node *s_i_* is:(24)$$p_{R}^{\prime \prime }=\frac{p_{R}^{\prime }}{\pi R_{0}^{2}}$$Substituting Equation (23) into Equation (24), we have:(25)$$p_{R}^{\prime \prime }=\frac{1}{\pi^{2}}\left(\frac{\pi }{4}-\frac{\lambda }{8}{\left(4-\lambda^{2}\right)}^{\frac{1}{2}}+\frac{\lambda^{2}\arccos\left(\frac{\lambda }{2}\right)}{2}-\frac{1}{16}\lambda^{3}{\left(4-\lambda^{2}\right)}^{\frac{1}{2}}\right)$$The degree of redundant coverage formed by *n* neighboring nodes centering this redundant node *s_i_* with sensing radius *R_a_* is as follows:(26)$$p_{R}=1-\left(1-p_{R}^{\prime \prime }\right)=1-{\left(1-\frac{1}{\pi^{2}}\left(\frac{\pi }{4}-\frac{1}{8}{\left(4-\lambda^{2}\right)}^{\frac{1}{2}}+\frac{\lambda^{2}\arccos\left(\frac{\lambda }{2}\right)}{2}-\frac{1}{16}\lambda^{3}{\left(4-\lambda^{2}\right)}^{\frac{1}{2}}\right)\right)}^{n}$$This completes the proof. □

### 3.4. Expectation of First Covarage

The expectation of the coverage quality within the monitoring area is presented in Theorem 1 and the degree of redundant coverage is given in Theorem 2. Now we focus on the expectation of first coverage when the sensor nodes firstly covered the path of the target node which is moving. After acquiring this expectation, this node could utilize the awakening mechanism to make the subsequent nodes accomplish the coverage task over the moving target node with the same or approximate expectation value. Therefore, we introduce Theorems 3, 4, and 5.

**Theorem** **3.**
*The expectation of first coverage when the first time that a sensor node covered the moving target node is E(C_f_) = [1 − (1 − C_p_)^Q^]/C_p_ where Q is the maximal number of transferring for the moving target node within unit time.*


**Proof.** We assume the random variable *X* is the number of transferring within unit time and *X* ∈ [1, 2, …, *Q*]. When *X* = *m*, the target node which is moving is not covered for the first *Q* − 1 times if and only if *m* < *Q* − 1, where the probability density function for the distribution of *X* is:(27)$$p\left(X=x\right)=\left\{ \begin{array}{ll} C_{p}{\left(1-C_{p}\right)}^{k-1} & k=1,2,\cdots Q-1\\ {\left(1-C_{p}\right)}^{Q-1} & k=Q \end{array}\right.$$Then we derive the expectation as follows:(28)$$E\left(C_{f}\right)=\sum_{k=1}^{Q-1}kC_{p}{\left(1-C_{p}\right)}^{k-1}+Q{\left(1-C_{p}\right)}^{Q-1}$$Setting $q=1-C_{p}$, $Y=\sum_{k=1}^{Q-1}k{\left(1-C_{p}\right)}^{k-1}$, then:(29)$$Y=\sum_{k=1}^{Q-1}kq^{k-1}$$Multiply *q* at both sides of the equation and further simplify the equation, we have:(30)$$Y=\frac{1-q^{Q-1}}{{\left(1-q\right)}^{2}}-\frac{\left(Q-1\right)q^{Q-1}}{1-q}$$Substituting $q=1-C_{p}$ into Equation (30), we have:(31)$$Y=\frac{1-{\left(1-C_{p}\right)}^{Q}}{C_{p}{}^{2}}-\frac{Q{\left(1-C_{p}\right)}^{Q-1}}{C_{p}}$$Substituting Equation (31) into Equation (28), we have:(32)$$E\left(C_{f}\right)=\frac{1-{\left(1-C_{p}\right)}^{Q-1}+C_{p}{\left(1-C_{p}\right)}^{Q-1}}{C_{p}}=\frac{1-{\left(1-C_{p}\right)}^{Q}}{C_{p}}$$This completes the proof. □

When one moving target node moves across the monitoring area, we perform data fitting on the path of the moving target node and give the function of the node trajectory.
(33)$$f_{1}(x)=\delta_{n}x^{n}+\delta_{n-1}x^{n-1}+\delta_{n-2}x^{n-2}+\cdots +\delta_{1}x^{1}+\xi$$

Set the range of value for *δ* to be *δ* ∈ [0, 1]. In order to calculate the size of the shadowed area, we establish the 2-dimension panel and mark the Y-axis and the X-axis. The area which is shadowed and the partial shape between sensor nodes *a* and *b*, as well as the node *p*, which is a sensor node at the lower-left corner, are shown in Figure 3.

In Figure 3, while the intersection point is the origin, we make Y-axis and X-axis to move through the centers of sensor nodes *b* and *a*. Then we connect sensor nodes *a* and *b* and assume that the cutting point is *c*. The trajectory of the target node which is moving and the X-axis intersect at point *e*. The trajectory of the target node which is moving and the Y-axis intersect at point *d.* The coordinate of node *a* is (*x_a_*, 0) while that for node *b* is (0, *y_b_*). Since the point *c* is the cutting point between node *a* and node *b*, the coordinate of node *c* is (0.5*x*, 0.5*y*) while those for node *e* and node *d* are (*x_a_* + *r*, 0) and (0, *y_b_* + *r*), respectively. The size of the shadowed area in Figure 3 is equal to the area surrounded by axis-X and axis-Y minus the size of a lower corner and one circle. As for the lower corner size, we can also perform the data fitting to give an equation. Set *b*∈ [0, 1], the equation for the curve is:(34)$$f_{2}(x)=b_{n}x^{n}+b_{n-1}x^{n-1}+b_{n-2}x^{n-2}+\cdots +b_{1}x^{1}+\xi$$

And the size of the shadowed area is:(35)$$S=\int_{0}^{x_{a}+r}f_{1}(x)dx-\int_{0}^{x_{a}-r}f_{2}(x)dx-\pi r^{2}$$

Substituting Equations (33) and (34) into (35), we have:(36)$$S=\frac{1}{i+1}\sum_{i=1}^{n}\left[a_{i}{\left(x_{a}+r\right)}^{i+1}-b_{i}{\left(x_{a}-r\right)}^{i+1}\right]-\pi r^{2}$$

**Theorem** **4.**
*Without the loss of any generality, if a void exists and the g(x) and the void curves f(x) are consecutive and derivable, after that, the difference value between the sizes of the two voids is no smaller than the sum of the sizes of two externally tangent circle sectors.*


**Proof.** As shown in Figure 3, if we move the coordinate of the X-axis and the Y-axis to *O*’, we have new coordinate axes *X*’ and *Y*’. The sensor node *a* intersects with *Y*’ at *n* and *m* and the angle formed is *β*. Similarly, sensor node *b* intersects with the *X*’ axis at point *f* and *g* and the angle formed is *α*. Due to the existence of voids, Δ*S* = |*S*_1_ − *S*_2_ − *S_fan_*| ≥ 0. According to Equations (35) and (36), we have:(37)$$\Delta S=\frac{1}{i+1}\sum_{i=1}^{n}\left[a_{i}{\left(x_{a}+r\right)}^{i+1}-b_{i}{\left(x_{a}-r\right)}^{i+1}\right]-S_{fan}$$And since *S_fan_* = 2π*r*^2^ − [*r*^2^(*α* + *β*)/2], we substitute this into Equation (37) and have the following result:(38)$$\Delta S=\frac{1}{i+1}\sum_{i=1}^{n}\left[a_{i}{\left(x_{a}+r\right)}^{i+1}-b_{i}{\left(x_{a}-r\right)}^{i+1}\right]-S_{fan}=\frac{1}{i+1}\sum_{i=1}^{n}\left[a_{i}{\left(x_{a}+r\right)}^{i+1}-b_{i}{\left(x_{a}-r\right)}^{i+1}\right]-2\pi r^{2}-\frac{1}{2}r^{2}\left(\alpha +\beta \right)$$Due to the existence of voids, Δ*S* ≥ 0. Therefore,
(39)$$\frac{1}{i+1}\sum_{i=1}^{n}\left[a_{i}{\left(x_{a}+r\right)}^{i+1}-b_{i}{\left(x_{a}-r\right)}^{i+1}\right]\geq 2\pi r^{2}+\frac{1}{2}r^{2}\left(\alpha +\beta \right)$$And the proof is completed. □

In theorem 4, we can obtain that when {0 ≤ *α* ≤ π}∩{0 ≤ *β* ≤ π}, the area of void that needs to be filled is at least 2π*r*^2^. If the angles *α* and *β* lies between [π, 2π], then the area of void that needs to be filled is at least π*r*^2^.

**Theorem** **5.**
*Without any generality loss, if void exists, i.e., x ≥ 0 and the void is consecutive and derivable in [0, a], the trajectory function of the moving target node satisfies $f_{n}\left(x\right)=\int_{0}^{x}f_{n-1}\left(x\right)dx$, then the series $\sum_{n=0}^{\infty }f_{n}\left(x\right)$ is definitely existing and convergent in [0, a].*


**Proof.** ince *x* ≥ 0 and consecutive and derivable on [0, *a*], |*f*_0_(*x*)| is consecutive on [0, *a*]. Therefore, |*f*_0_(*x*)| has a maximum on [0, *a*], which is denoted as *M*. That is, *M* is the maximum value in the range and it is equal to $M=\underset{0\leq x\leq a}{max}\left|f_{0}\left(x\right)\right|$. Since:(40)$$f_{n}\left(x\right)=\int_{0}^{x}f_{n-1}\left(x\right)dx=\int_{0}^{x}\left[\int_{0}^{x}f_{n-2}\left(x\right)dx\right]dx=\cdots =\underset{n}{\underset{︸}{\int_{0}^{x}\cdots }}{\int_{0}^{x}f_{0}\left(x\right)\left(dx\right)}^{n}$$Since |*f*_0_(*x*)| is consecutive on [0, *a*], |*f*_0_(*x*)| has a maximum value *M* on [0, *a*]. Therefore:(41)$$\left|f_{n}\left(x\right)\right|\leq \underset{n}{\underset{︸}{\int_{0}^{x}\cdots }}\int_{0}^{x}\left|f_{0}\left(x\right)\right|dx\leq \underset{n}{\underset{︸}{\int_{0}^{x}\cdots }}\int_{0}^{x}Mdx=\frac{M}{n!}x^{n}$$Since Equation (41) is convergent on *x* ∈ [0, 1], $\sum_{n=0}^{\infty }\left|f_{n}\left(x\right)\right|$ is convergent and $\sum_{n=0}^{\infty }f_{n}\left(x\right)$ is definitely convergent. This proof is completed. □

## 4. Realization of CMTN-SP Algorithm

### 4.1. Algorithm Ideas

The Coverage Control Algorithm for Moving Target Nodes based on Sensing Probability (CMTN-SP) algorithm employs the dynamic clustering mechanism to set some nodes as the cluster heads and the cluster heads could form one backbone network with self-processing ability and relaying data. For other non-backbone network nodes, they can switch into sleep state by the node state transferring mechanism to save the energy consumption [32,33,34,35]. The organization forms of clusters are mainly characterized by the fact that each cluster is composed of one cluster head and several cluster members. The cluster members could communicate directly or indirectly with the cluster head. When there is a common member between different clusters, the communication can be accomplished via the gateway node [36,37,38,39]. The cluster head is mainly in charge of transmitting and receiving the data information within the cluster, data fusion and data storage. The data communication among the cluster members or between the cluster member and the cluster head can be performed in a time division duplexing manner, to avoid the collision of data packets and packet losses. On the other hand, the nodes not in the communication are switched to sleep state to save the energy consumption. When the cluster heads receive the response messages from the cluster members, the cluster heads assign the working time slots to the cluster member node, which accomplishes the time synchronization.

The proposed algorithm is carried out mainly in three periods: (1) formation of the clusters, (2) election of the cluster heads, and (3) dynamic changes. Firstly, when the other nodes receive the request frames from the node with higher remaining energy, they calculate the distance to this node according to the location information and perform comparison on distance in the local topology structured list. If this distance is smaller than the threshold, the topology structured list bit is changed to 1. Otherwise, it is changed to 0. After one or more time rounds, the nodes satisfying the conditions are put into a set, which forms the cluster. Secondly, in the cluster head election period, the nodes with more remaining energy are mainly in charge of electing the node with the maximal coverage expectation as the head of the cluster. The cluster head which is elected sends a request frame to the cluster members in a flooding schedule, which contains the ID and location information of this cluster head. This request also serves as a declaration of election. For the convenience of management, this cluster head establish a dynamic link to store the cluster members the remaining energy, the distances among cluster members, the ID of the members and the cluster members sensing radius. Thirdly, since the cluster head is in charge of many tasks, the energy consumption is quick for this node. Furthermore, the energy imbalance among the cluster members leads to the problem that the total energy of this cluster could not sustain the energy consumption after many rounds of tasks. Therefore, in order to guarantee the subsequent work, we need to perform another round of division and integration for the clusters with lower total energy. In the dynamic cluster partition process, we need to consider the node remaining energy in the local topology structured list, expectation of the coverage and the distance among nodes. Then we choose the appropriate node as the cluster head. After that, according to the distance among nodes, we search for the nodes with bit 1 in the local topology structured list and form a cluster with these nodes.

When the time duration of an event is long, the cluster energy would quickly run out if the cluster head is fixed [40,41,42,43]. Therefore, we have to perform the switching of the cluster head and in this switching process, we need to consider the round of data transmission, node HTS (Hop To mobile Sink) value and the remaining energy to choose a cluster head. Therefore, we can balance the responsibility of the cluster head and save the energy consumption of cluster reformation. When the best candidate node does not have enough energy to perform the cluster reformation, the current cluster head sends a PCM (Position Calculation Message, binary tuple <Type, E_ID>) message to the MS (Mobile Sink), which aims to timely urge the MS to perform the quasi-optimal location computation, update the network routing structure, and reduce the event field data loss caused by the absence of cluster heads.

### 4.2. Realization of Algorithm

**Step 1:** Calculate the expectation of coverage and sensing properties for all the nodes and store in the local topology structured list.

**Step 2:** Search for the node in the topology structured list satisfying the condition as the cluster head.

**Step 3:** The cluster heads broadcast the request frame in a flooding schedule that includes this cluster head ID.

**Step 4:** After the cluster members receive the request frame, they send the response message to the cluster head, containing the cluster member’s ID, remaining energy, sensing property, etc. These messages are stored in the local topology structured list of the cluster head.

**Step 5:** The nodes not in the local topology structured list of the cluster head are switched into sleep mode.

**Step 6:** After many unit rounds, perform the comparison on the data stored in the local topology structured list of the cluster head. Select the nodes satisfying the condition as the new cluster head and turn to Step 3.

**Step 7:** When the total energy of this cluster is below the threshold *ε*, this cluster is dismissed.

### 4.3. Algorithm Complexity

We proposed a distributed algorithm in this work and the complexity of this algorithm includes the time complexity and spatial complexity. The number of nodes transmitting data within the cluster is *k*, the total quantity of the nodes is *N*, and *k* ≤ *N*. If all the nodes transmitting data share the same node energy and are time synchronized, the complexity of the time of the CMTN-SP algorithm is *O*(*k*) at its best. Otherwise, the time complexity is *O*(*k*log*k*). In terms of the spatial complexity, since the number of transmitting-data nodes *k* is determined in advance, the time complexity is therefore *O*(*k*).

## 5. System Evaluation

For verifying the stability and the effectiveness of this algorithm, we employ the platform, MATLAB 7.0, to carry out simulations and make comparisons with the PDA [22], DAT [24] and VLAD [25] algorithms in terms of the coverage rate of the network, network lifetime, and the quantity of survived nodes. The performances are listed in Table 1.

The network coverage performances versus the number of sensor nodes under different monitoring area are illustrated with *δ* = 0.3 and *δ* = 0.8 form Figure 4, Figure 5 and Figure 6, respectively. It is shown in the figures that the network coverage rate increases for all the four algorithms with the number of sensor nodes. Generally, the proposed algorithm outperforms the other three algorithms in terms of the network coverage rate. It is shown in Figure 4 that when the number of nodes is equal to 200, the coverage rate of the proposed algorithm is 80.2% and 75.4%, while those for the other algorithms are 72.1%, 69.95%, and 68.01%, which is below the performance of the proposed algorithm. It is shown in Figure 5 that when the monitoring area is 400 × 400 m^2^ and the number of sensor nodes is 300, the proposed algorithm shows a more prominent increase in network coverage rate than the others. The main reason behind this is that we adopt the controllable parameter *λ* to adjust the coverage rate within the entire monitoring area, so that we can adjust to the changing coverage rate during the consecutive coverage over the moving target node. The network coverage rate is improved by 16.32% on average. The analysis process in Figure 6 and Figure 7 is similar to that in Figure 4 and Figure 5. The comparison on the network coverage rate with *δ* = 0.3 and *δ* = 0.8 is also presented in Figure 8 and Figure 9 for the moving target node. We can see that with an increasing number of moving target node, the network coverage rate decreases for all the four algorithms. But the proposed algorithm shows the least magnitude of decrease, which does not influence the coverage effect over the entire network. The main reason behind this is that the controllable parameter could maintain the network coverage within a certain range, which guarantees the robustness and scalability for the proposed algorithm.

Figure 10, Figure 11, Figure 12 and Figure 13 give the network lifetime performance versus the number of sensor nodes with *δ* = 0.3 and *δ* = 0.8. It is shown from the figures that with an increasing number of sensor nodes, the network lifetime is prolonged for all the four algorithms. In Figure 10, when the number of sensor nodes is 250, the network lifetime of the proposed algorithm is 450 s and 420 s, while those for the other algorithms are 410 s, 305 s and 295 s. In Figure 11, when the number of nodes is 500, the network lifetime of the proposed algorithm is 520 s and 400 s while those for the other algorithms are 380 s, 340 s, and 240 s. The main reason behind this is that the other three algorithms employ the centralized coverage while we employ the distributed coverage. Meanwhile, the other three algorithms do not consider the transformation process among different nodes and they require more energy consumption for the network. The proposed algorithm increases the network lifetime by 13.07% compared with the other three algorithms. The network lifetime versus the number of moving target nodes with *δ* = 0.3 and *δ* = 0.8 are shown in Figure 14 and Figure 15. It is shown that with an increasing number of moving target nodes, the network lifetime of all the four algorithms are decreased. We take Figure 14 as an example. When the number of moving target nodes is 6, the magnitude of decrease is larger for the other three algorithms. The main reason behind this is that the other three algorithms employs the centralized coverage scheme, without considering the coordination among different nodes, which leads to the rapid exhaustion of these three algorithms and a shorter network lifetime compared with the proposed algorithm.

Form Figure 16, Figure 17, Figure 18 and Figure 19 give the number of working nodes versus the total number of nodes with *δ* = 0.3 and *δ* = 0.8. It is shown in the figures that with the increase in the total number of nodes, the number of working nodes remain relatively steady for all the four algorithms. However, the required number of working nodes of the proposed CMTN-SP algorithm is smaller than the other algorithms, which is 8.26%. The main reason behind this is that the proposed algorithm performs consecutive coverage over the focused moving target node while the other nodes not participating in the coverage or the redundant nodes are switched into sleep state or overhearing state to prolong the network lifetime. By contrast, the other three algorithms perform the consecutive coverage over the entire monitoring area and only parts of the nodes are switched into sleep state or overhearing state.

## 6. Conclusions

This paper first analyzed and investigated the coverage condition of the moving target node. Based on this, a Coverage Control Algorithm for Moving Target Nodes Based on Sensing Probability Model was proposed. According to the normal distribution function, this proposed algorithm gave the derivation method for the network coverage quality and the minimal number of nodes. Then we analyzed and investigated the generation of the redundant data and further presented the calculation method for the degree of redundant coverage. In terms of the realization of this algorithm, we gave the key ideas and the detailed description of the CMTN-SP algorithm, as well as the complexity of this algorithm. Finally, we verified via simulations the network coverage rate, network lifetime and the number of working nodes for the proposed algorithm, in comparison with the other three algorithms. The simulation results proved the effectiveness and the stability of the proposed algorithm.

## Figures and Tables

**Figure 1 sensors-19-00257-f001:**
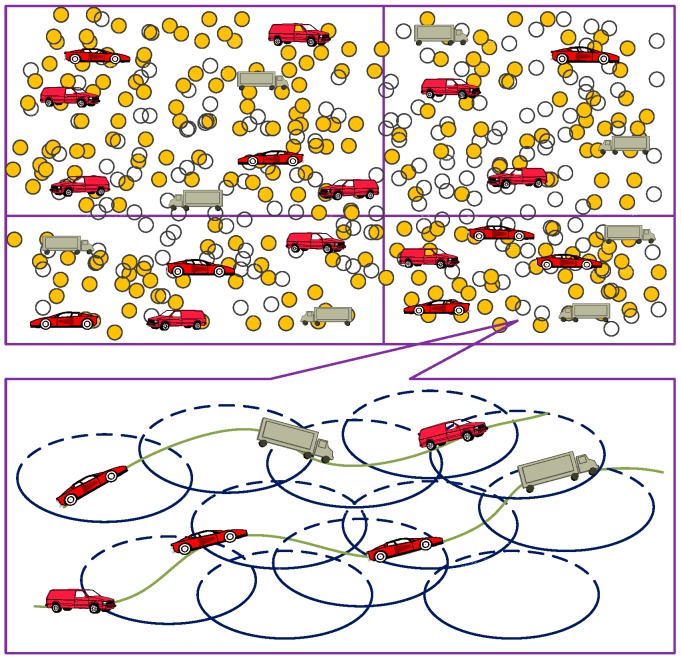
Consecutive coverage over the moving target node.

**Figure 2 sensors-19-00257-f002:**
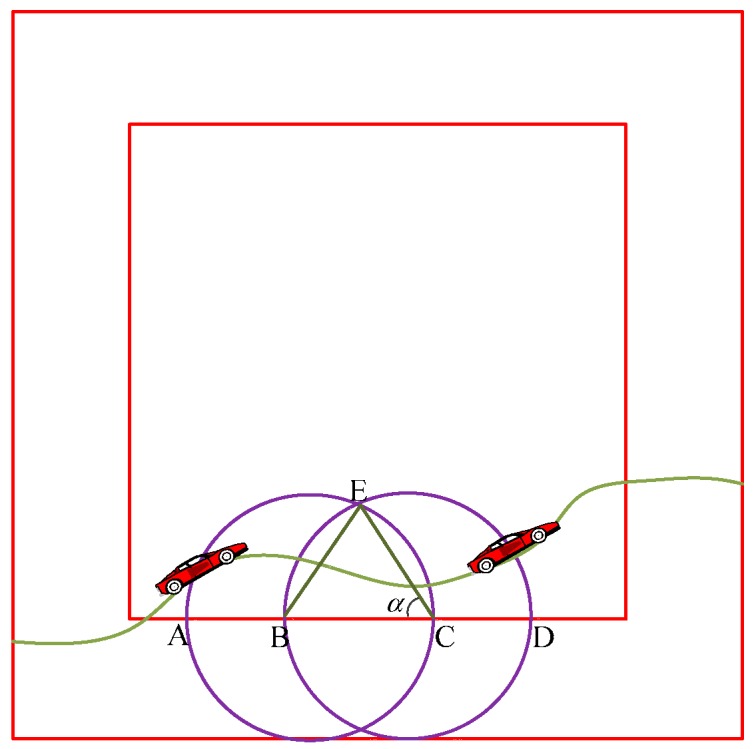
Illustration of the redundant coverage.

**Figure 3 sensors-19-00257-f003:**
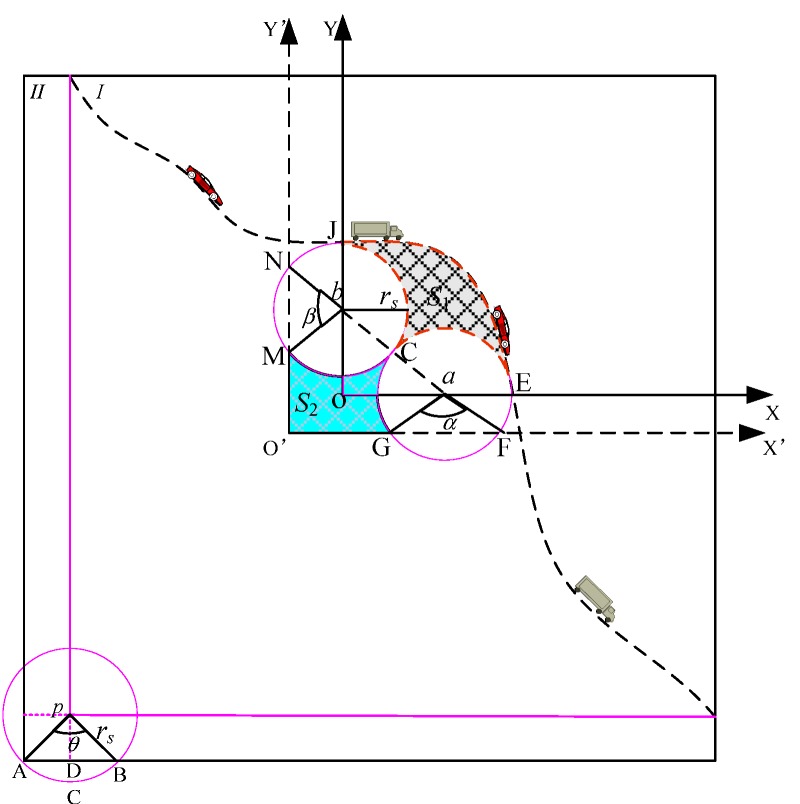
The breakdown illustration of the coverage.

**Figure 4 sensors-19-00257-f004:**
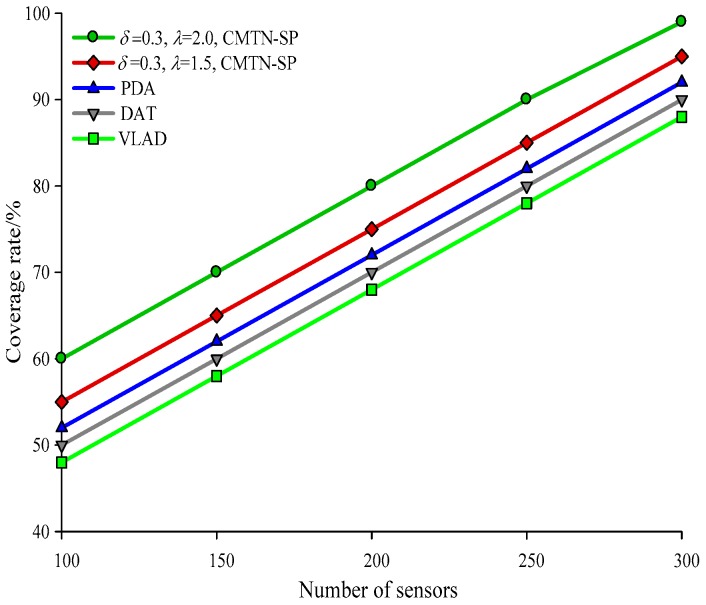
Networks coverage with 300 × 300 m^2^, *δ* = 0.3.

**Figure 5 sensors-19-00257-f005:**
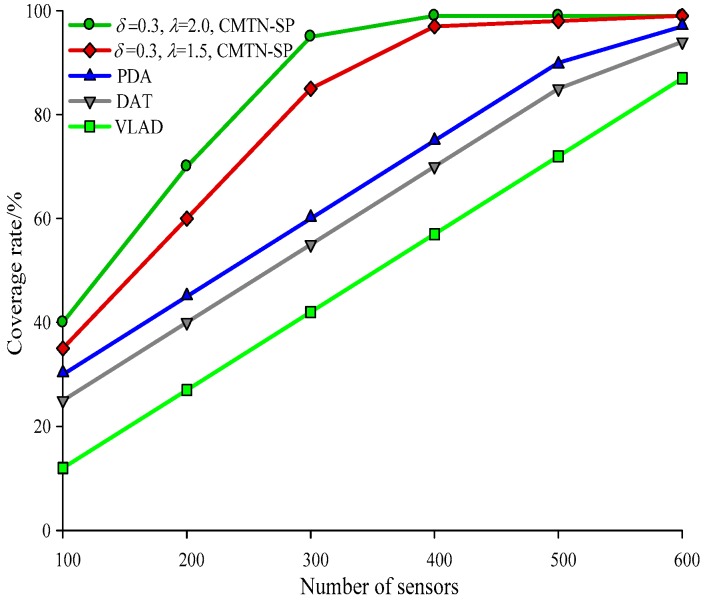
Networks coverage with 400 × 400 m^2^, *δ* = 0.3.

**Figure 6 sensors-19-00257-f006:**
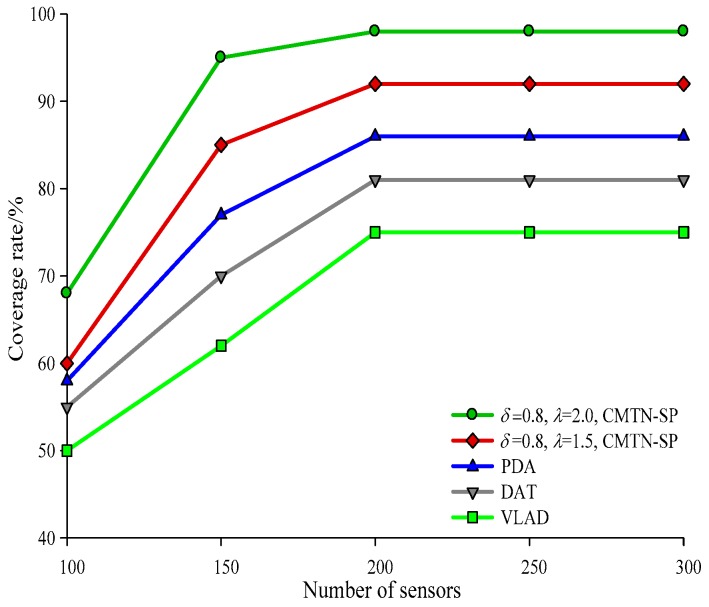
Networks coverage with 300 × 300 m^2^, *δ* = 0.8.

**Figure 7 sensors-19-00257-f007:**
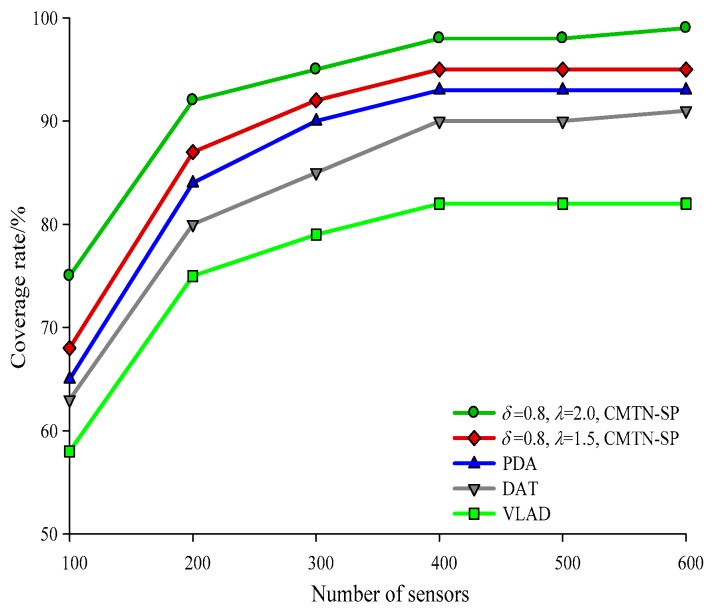
Networks coverage with 400 × 400 m^2^, *δ* = 0.8.

**Figure 8 sensors-19-00257-f008:**
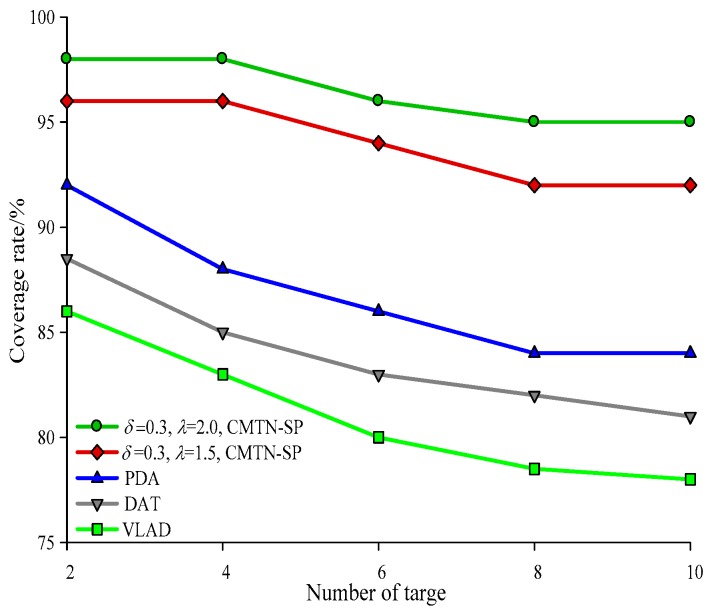
Networks coverage with 300 × 300 m^2^, *δ* = 0.3.

**Figure 9 sensors-19-00257-f009:**
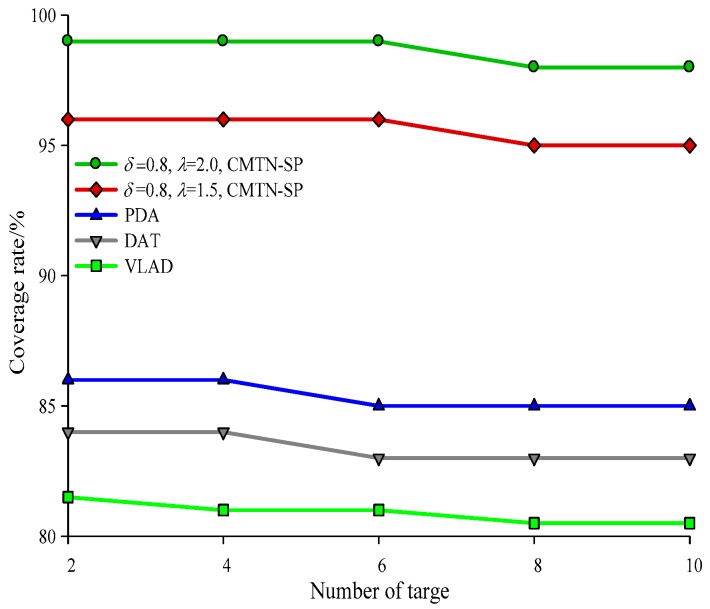
Networks coverage with 300 × 300 m^2^, *δ* = 0.8.

**Figure 10 sensors-19-00257-f010:**
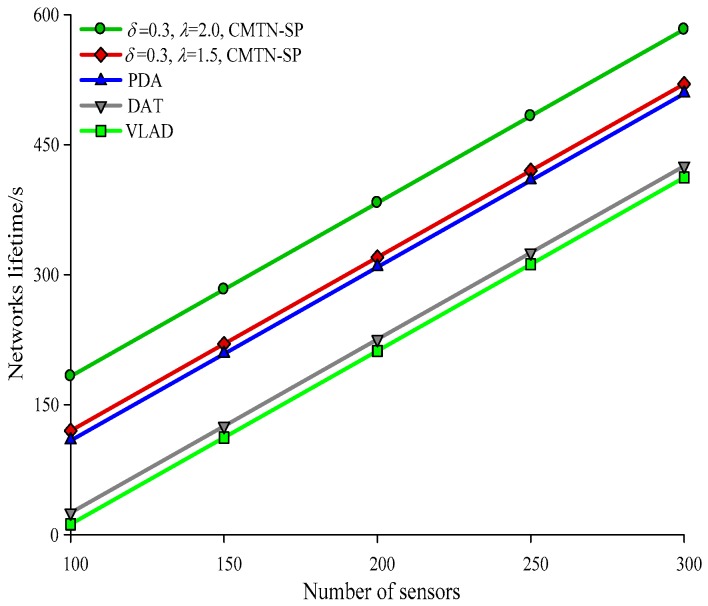
Networks lifetime with 300 × 300 m^2^, *δ* = 0.3.

**Figure 11 sensors-19-00257-f011:**
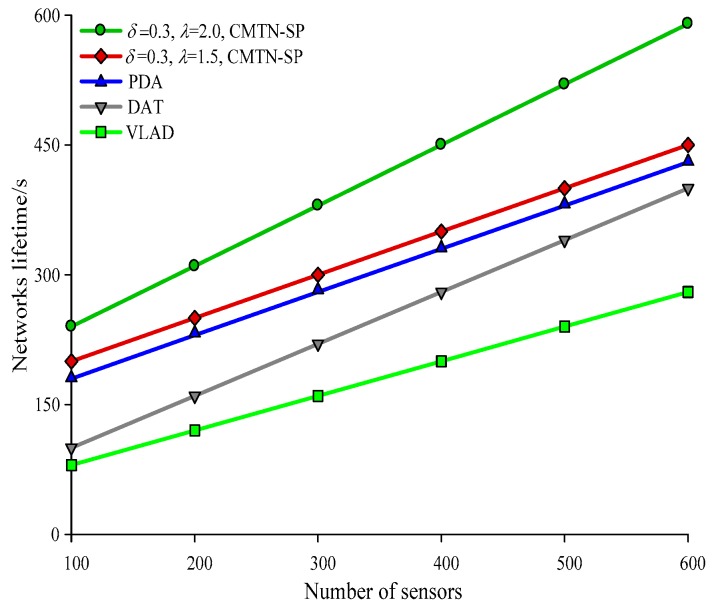
Networks lifetime with 400 × 400 m^2^, *δ* = 0.3.

**Figure 12 sensors-19-00257-f012:**
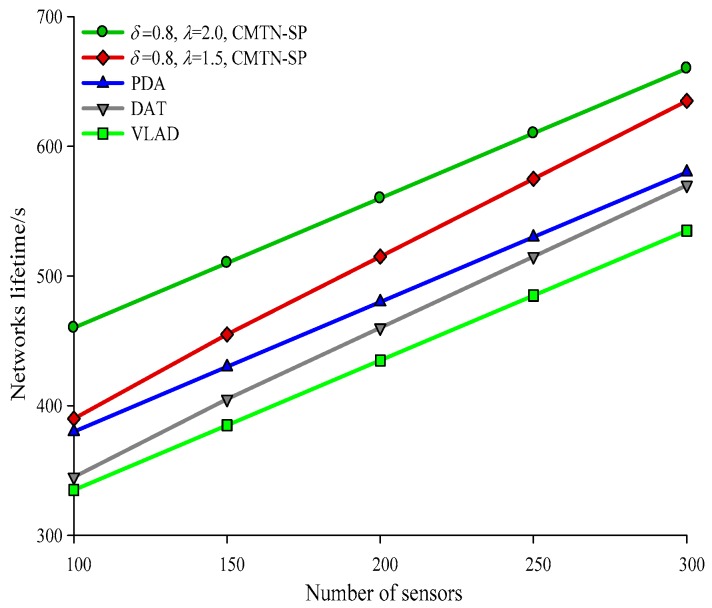
Networks lifetime with 300 × 300 m^2^, *δ* = 0.8.

**Figure 13 sensors-19-00257-f013:**
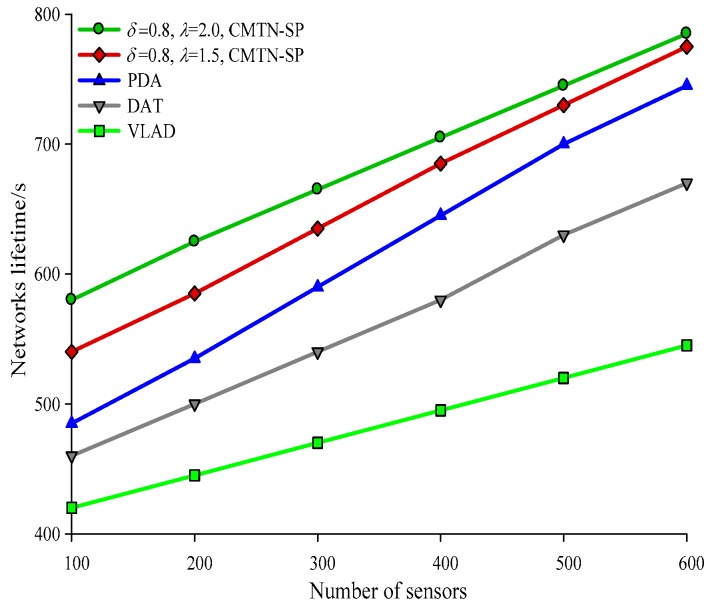
Networks lifetime with 400 × 400 m^2^, *δ* = 0.8.

**Figure 14 sensors-19-00257-f014:**
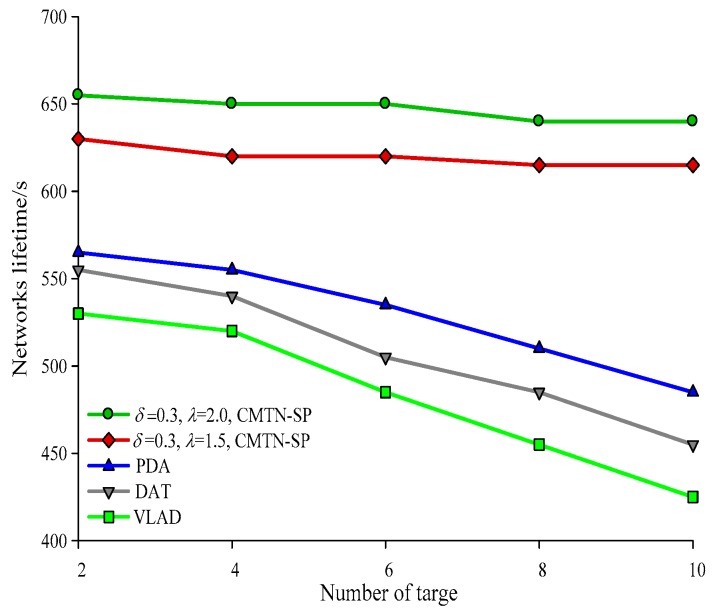
Networks lifetime with 400 × 400 m^2^, *δ* = 0.3.

**Figure 15 sensors-19-00257-f015:**
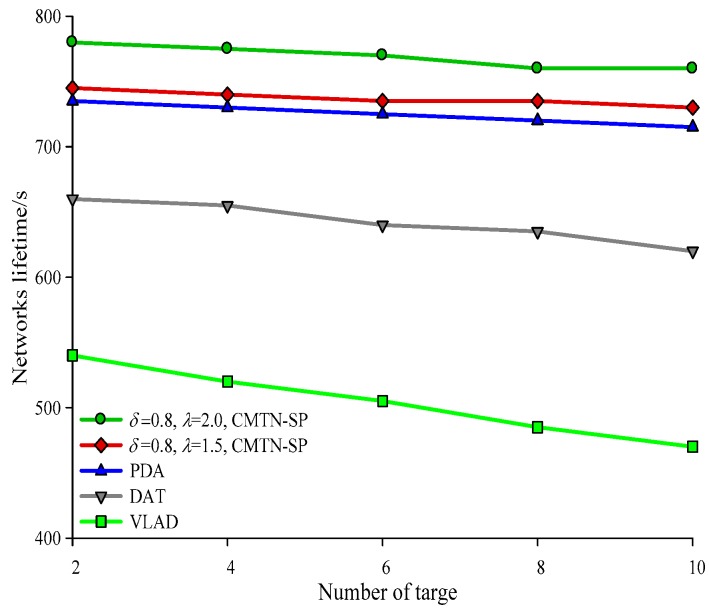
Networks lifetime with 400 × 400 m^2^, *δ* = 0.8.

**Figure 16 sensors-19-00257-f016:**
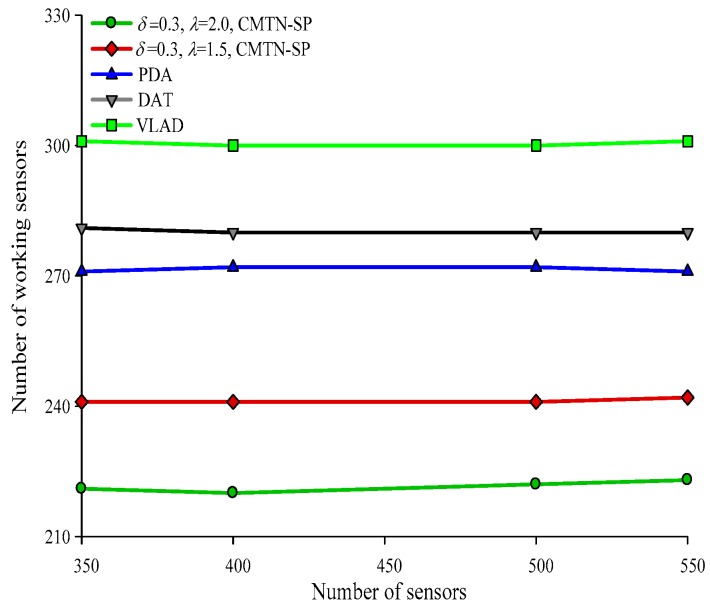
The number of working nodes versus the number of nodes with 300 × 300 m^2^, *δ* = 0.3.

**Figure 17 sensors-19-00257-f017:**
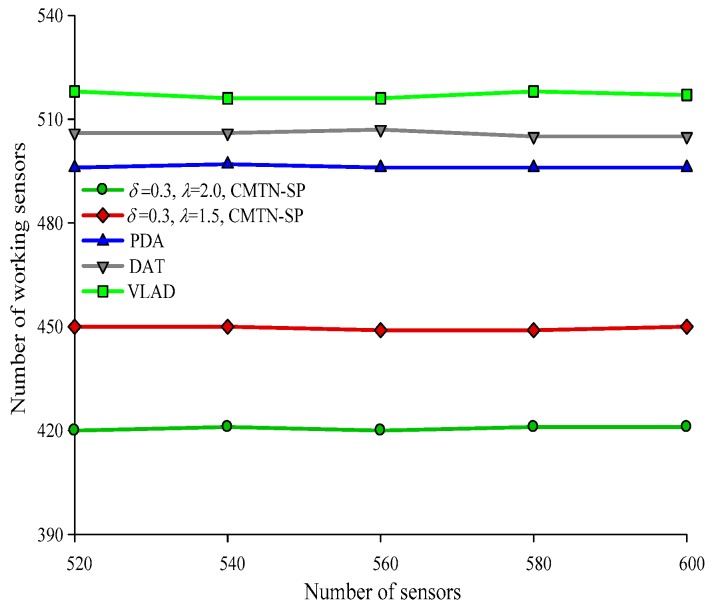
The number of working nodes versus the number of nodes with 400 × 400 m^2^, *δ* = 0.3.

**Figure 18 sensors-19-00257-f018:**
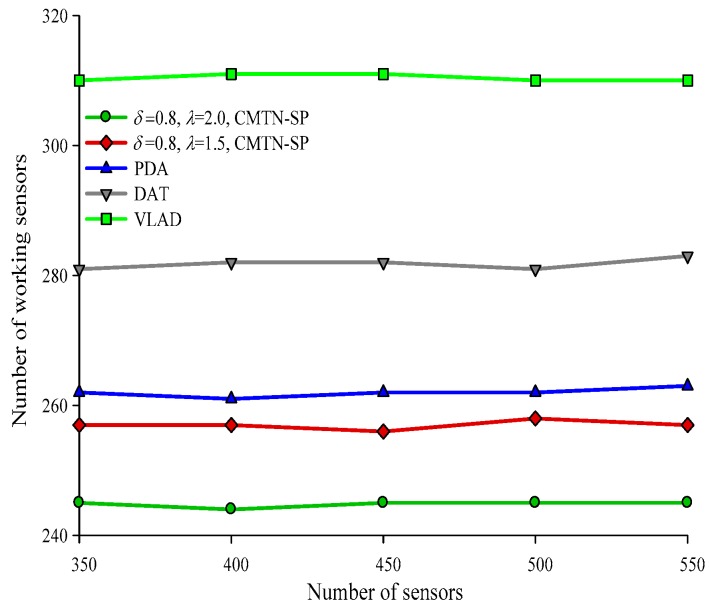
The number of working nodes versus the number of nodes with 300 × 300 m^2^, *δ* = 0.8.

**Figure 19 sensors-19-00257-f019:**
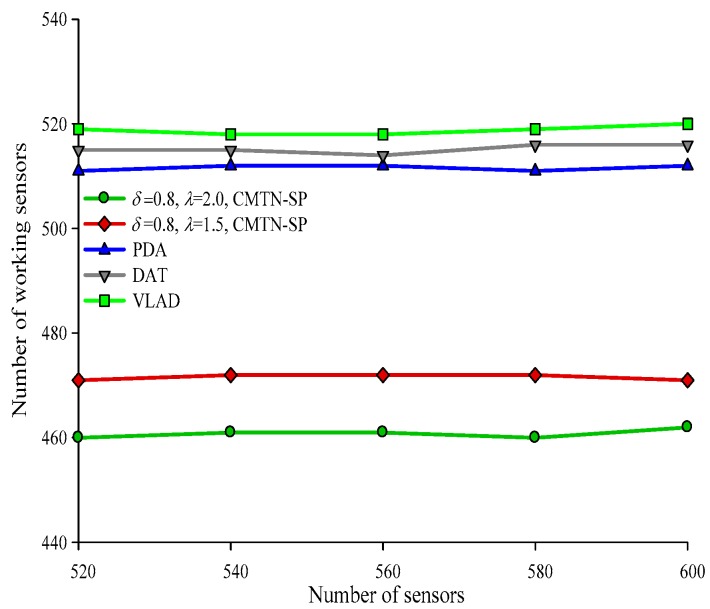
The number of working nodes versus the number of nodes with 400 × 400 m^2^, *δ* = 0.8.

**Table 1 sensors-19-00257-t001:** Performance comparison results.

Simulation Parameter	Value	Simulation Parameter	Value
Monitoring area	300 × 300 m^2^	time	200 s
Monitoring area	400 × 400 m^2^	*R_c_*	20 m
*R_s_*	10 m	*E* _R-elec_	50 J/b
Initial energy	10 J	*E* _T-elec_	50 J/b
Number of sensors	800	*ε* _fs_	10 (J/b)/m^2^

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
