# Peer review of "CMTN-SP: A Novel Coverage-Control Algorithm for Moving-Target Nodes Based on Sensing Probability Model in Sensor Networks"

_sensors, 2019, doi:10.3390/s19020257_

Reviewer 1 Report

Detailed comments are as follows:

1.       The Abstract is too wordy, especially, there is the repetition of the concepts between lines 22-29.

2.       You must first explicitly define and differentiate the coverage area, monitoring area, and the sensing area.

3.       What is the example of circular sensing that has practically used in WSN?

4.       The proposed model in the manuscript revolves around the sensors and their efficiency, however, authors did not discuss what type of sensors are could be used in the example model scenario? What are the basic characteristics of the sensor nodes in the network?

5.        Review the proof between 170-200.  Authors missed a lot of details. It is not just to give long expressions, but the expressions with meaning.

6.       The illustration in Fig. 2 is vague, poorly discussed, and cannot properly represent the concept of redundant coverage. The concept of redundant coverage should be shown using different nodes explicitly showing the coverage area and sensing area.

7.       How effectively you can simulate the algorithm without a proper explanation of the parameter and the environment. Authors must simulate their algorithm in a well-investigated simulator than the Matlab.

8.       The algorithm steps must be implemented in the network simulators.

9.       What is the control overhead? How much energy is consumed by the control overhead?

10.   Why is the packet drop ratio (the data and control packets) in the network?

Typos:

a.       Kindly replace the word “casting” at line 36, which is inappropriate word choice in that context.

b.      Revise punctuation in Line 73.

Author Response

We would like to thank the reviewers for the comments on this manuscript, which helps us to greatly improve the quality of this wok. We would like to express our sincere gratitude to the reviewers’ hard work. We reply the comments in attached file 1.

Reviewer 2 Report

The issue of sensor network's coverage had been done by researchers many years ago e.g. in the following articles. It is not clear the motivation and the difference of this paper compared to conventional works. By the way, the authors should take into account the shadowing probability when computing the coverage of sensors. 

https://link.springer.com/chapter/10.1007/11599463_69

https://ieeexplore.ieee.org/document/6217961

https://ieeexplore.ieee.org/document/4810014

Author Response

We would like to thank the reviewers for the comments on this manuscript, which helps us to greatly improve the quality of this wok. We would like to express our sincere gratitude to the reviewers’ hard work. We reply the comments in attached file 2.

Round  2

Reviewer 1 Report

The authors could not properly answer the following comments that were asked in the previous review:

You must first explicitly define and differentiate the coverage area, monitoring sensing area.

(The definition of sensing area is not the intersection of the coverage area of the sensor nodes and its neighbors. It is the area where a sensor can detect/sense the physical phenomenon. Basically, it is the property of the sensor attached with the wireless communication circuitry, which is collectively called the sensor node.)

Usually, the coverage area of a WSN node is the farthest range where it can successfully communicate with the other nodes in the network.

The authors replied to the comment related to the practical example of the circular sensing as "circular sensing area with the sensor at the center. This is not the example of the circular sensing. Based on the reply, can authors please explain the novel phenomenon that they represented:

   "What is the relationship between transmission/transmitting power and sensing range?" 

The authors were asked about what type of sensors they considered or assumed in the study and they replied with the subsections of the WSN node. I think they failed to understand the comment. My main question was either they assume MicaZ with thermisters, TelosB with photosensor, etc. In the next part of the comment, I asked about the main characteristics of the sensor nodes, and the authors' response just partially covers that part of the comment.

In my opinion, authors need to reconsider the concept of redundant coverage and properly present in Fig. 2. It was not asked to show many sensors and draw many overlapping circles, but to properly DEFINE and ILLUSTRATE the terminology.

I can understand the limitations of the Matlab implementation, however, your answer that every node consumes the same amount of energy per second prompted a new ambiguity. If all nodes have the same amount of initial energy and they consume the same amount of energy per second, then they must deplete their energy at the same time. I know that there are other events that take place i.e., sensing and sampling intervals, discovery intervals, transmission overhead, node density, etc. however, the authors simplified all that phenomenon with a simple reply that each node consumer same amount of energy per second!

Author Response

Response to the comments of reviewers(ID:Sensors-410611)

We would like to thank the reviewers for the comments on this manuscript, which helps us to greatly improve the quality of this wok. We would like to express our sincere gratitude to the reviewers’ hard work.

Review comments: (round 2)

1. The authors could not properly answer the following comments that were asked in the previous review:

You must first explicitly define and differentiate the coverage area, monitoring sensing area.

(The definition of sensing area is not the intersection of the coverage area of the sensor nodes and its neighbors. It is the area where a sensor can detect/sense the physical phenomenon. Basically, it is the property of the sensor attached with the wireless communication circuitry, which is collectively called the sensor node.)

Usually, the coverage area of a WSN node is the farthest range where it can successfully communicate with the other nodes in the network.

Reply: we remove the definition of monitoring area and sensing area as they are not suit to give it as a formally definition.

Definition 1. (coverage region): The coverage region is defined as the overall area covered by all the deployed sensor nodes.

The authors replied to the comment related to the practical example of the circular sensing as "circular sensing area with the sensor at the center. This is not the example of the circular sensing. Based on the reply, can authors please explain the novel phenomenon that they represented:

 "What is the relationship between transmission/transmitting power and sensing range?"

Reply: in fact, we use a 0-1 sensing model to represent a sensor’s sensing region. A point can be covered/sensed by a sensor if only the distance between the point and the sensor is less than sensor’s sensing radius.

For the relationship between transmission/transmitting power and sensing range, the larger transmission power, the larger sensor’s sensing range.

The authors were asked about what type of sensors they considered or assumed in the study and they replied with the subsections of the WSN node. I think they failed to understand the comment. My main question was either they assume MicaZ with thermisters, TelosB with photosensor, etc. In the next part of the comment, I asked about the main characteristics of the sensor nodes, and the authors' response just partially covers that part of the comment.

Reply: The high-speed visual sensors are adopted by this work. The main characteristics of this sensor are listed below. (1) Data collection unit. The data collection unit is mainly composed by the sensor and the analog/digit converter, where the data collection module is mainly in charge of collection the mobile data and the analog/digit converter is in charge of converting the collected analog signal into digital signal. (2) Control unit. The control unit is in charge of the running of the entire sensor node, which is normally composed of the micro-processor and the memory. The function of the micro-processor is to perform the real-time processing on the data collected by the node and received from other nodes. On the other hand, the function of memory is to store the temporary data collected by the node and received from other nodes. (3) Communication unit. The wireless communication unit is in charge of the communication between different sensor nodes, exchanging control messages and transcribing data. (4) Power unit. The power unit is in charge of powering the sensor nodes to guarantee the functioning. It is crucial to the secure and reliable working of the entire network.

In my opinion, authors need to reconsider the concept of redundant coverage and properly present in Fig. 2. It was not asked to show many sensors and draw many overlapping circles, but to properly DEFINE and ILLUSTRATE the terminology.

Reply:

In order to guarantee the consecutive coverage which is in the monitoring area over the moving target node and reduce the quantity of redundant nodes and redundant data, we analyze the degree of redundant coverage. We assume the monitoring area is a double square where the outer square is not the focusing area for the moving target node while the inner square is the focusing area for the target node which is moving while the inner square is the focusing area for the target node which is moving. Generally speaking, the existence of other redundant nodes is ignored for investigation convenience. The illustration of the degree of redundant coverage is shown in Fig. 2.

Figure 2. Illustration of the redundant coverage

Definition 7. (Redundant Node): When the threshold is higher than the Euclidean distance between two arbitrary nodes, this node is defined as the redundant node, i.e., d(si,sj)<e(l).

Definition 8. (Degree of Redundant Coverage): When node si and its neighboring node are redundant nodes, the degree of redundant coverage is defined as the ratio of its neighbor’s coverage area to its own coverage area.

I can understand the limitations of the Matlab implementation, however, your answer that every node consumes the same amount of energy per second prompted a new ambiguity. If all nodes have the same amount of initial energy and they consume the same amount of energy per second, then they must deplete their energy at the same time. I know that there are other events that take place i.e., sensing and sampling intervals, discovery intervals, transmission overhead, node density, etc. however, the authors simplified all that phenomenon with a simple reply that each node consumer same amount of energy per second!

Response:

This question is very good and practical. We choose a sensor, for example Mica2, as a real sensor model. We carry out simulation works on Matlab platform as we do not find a convenient platform to implement the simulation. Sure, the sensing model of a sensor is simple compared with its practical properties. We will continue to perform this work on our future work. Thanks again.

Reviewer 2 Report

The authors did not response to the comments of the reviewer properly. The reviewer would like to rephrase the comments in the last round. 

(1) The issue of sensor network's coverage had been done by researchers many years ago e.g. in the provided references. Although the authors attempt to include all the mentioned references in the revised manuscript, the paper did not clearly explain the differences between this paper and conventional works. 

(2) The reviewer mentioned in the last round of review that the authors must take into account the shadowing effect in link budget design of the sensor network as well as computation of the coverage. But the reviewer did not see any responses from the authors regarding this issue. 

Author Response

Response to the comments of reviewers(ID:Sensors-410611)

We would like to thank the reviewers for the comments on this manuscript, which helps us to greatly improve the quality of this wok. We would like to express our sincere gratitude to the reviewers’ hard work.

Review comments:(Round 2)

(1) The issue of sensor network's coverage had been done by researchers many years ago e.g. in the provided references. Although the authors attempt to include all the mentioned references in the revised manuscript, the paper did not clearly explain the differences between this paper and conventional works.

(2) The reviewer mentioned in the last round of review that the authors must take into account the shadowing effect in link budget design of the sensor network as well as computation of the coverage. But the reviewer did not see any responses from the authors regarding this issue.

Response:

(1)    Thanks for the review expert’s comment. The issue of coverage in WSN has been done by many researchers many years ago. We do this work based on our previous research result, and proposed a coverage control algorithm for moving target nodes. Sure, some of our work is based on the some others work, for example, we adapted a probabilistic coverage model [7]. And we consider a scenario of monitoring a moving target to solve the fully coverage problem. For the mentioned link budget design of WSN, we assume the link is fixed if the residual energy of sensor is large than the threshold value, for easing to implement in Matlab simulator.

(2)    For the listed 3 references [7,11,40], we address its work in section 2 with red color. The last reference (Thai M.T.; Wang F.; Du D.Z. Coverage problems in wireless sensor networks: designs and analysis. In proceedings of the 2010 International Conference on Networking, Sensing and Control (ICNSC), 2010, 495-500. http://www.public.asu.edu/~fwang25/papers/Coverage.pdf), proposed a connected coverage and connected k-coverage theorem analysis based on a necessary and sufficient condition for the complete coverage of a convex region to simply connectivity is R>=2r. Its focus is the relationships between coverage and connectivity, and it is not too close relations to this work.

Ref.[7] Liu, M.; Cao, J.N.; Lou, W.; Chen, L.J.; Li, X. Coverage analysis for wireless sensor networks. Proceedings of 1st International Conference on Mobile Ad-hoc and Sensor Networks, MSN 2005, Wuhan, China, Springer Verlag, 13-15 December, 2005; pp. 711-720. https://link.springer.com/chapter/10.1007/11599463_69

Ref.[11] Li, W.; Zhang, W. Coverage analysis and active scheme of wireless sensor networks. IET Wireless Sensor Systems, 2012, 2, 86-91. https://ieeexplore.ieee.org/document/6217961

Ref.[40] Pudasainiy, S.; Mohx, S.; Shinz, S. Stochastic coverage analysis of wireless sensor networks with hybrid sensing model. Proceedings of 11th International Conference on Advanced Communication Technology, ICACT 2009, Phoenix Park Korea, IEEE Press, 15-18 February, 2009; pp. 549-553. https://ieeexplore.ieee. org/document/4810014

Round  3

Reviewer 2 Report

The authors have resolved the comments of the reviewer. The paper can be published as it is.